# Declining Growth Response of Siberian Spruce to Climate Variability on the Taiga–Tundra Border in the Putorana Mountains (Northwest Siberia)

Peter Fleischer [1,*], Viliam Pichler [1], Ján Merganič [1], Erika Gömöryová [1] 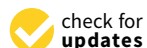, Marián Homolák [1] and Peter Fleischer, Jr. [1,2]

1 Faculty of Forestry, Technical University in Zvolen, T. G. Masaryka, 960 01 Zvolen, Slovakia; pichler@tuzvo.sk (V.P.); merganic@tuzvo.sk (J.M.); gomoryova@tuzvo.sk (E.G.); marian.homolak@tuzvo.sk (M.H.); xfleischer@is.tuzvo.sk (P.F.J.)
2 Institute of Forest Ecology, Slovak Academy of Sciences, 960 01 Zvolen, Slovakia
* Correspondence: yfleischer@is.tuzvo.sk

**Abstract:** Global warming is most pronounced at high latitudes where temperatures increase twice as fast as the global average. Boreal forest growth is generally limited by low temperatures, so elevated temperature is supposed to enhance biomass production and carbon sequestration. A large amount of evidence has recently shown inconsistent responses of tree growth derived from annual tree rings to increasing temperature. We studied Siberian spruce growth in the remote and isolated Putorana Mts, Western Siberia in populations at its natural distribution limit. Tree ring cores were sampled along vertical transect in 100, 200 and 350 m a.s.l. as the aim was to identify the tree growth rate at different altitudes. Detailed sampling site descriptions served to identify possible factors controlling the growth rate in extremely heterogeneous environments. Monthly climate data for the period 1900–2020 were extracted from the gridded CRU database. Tree ring chronologies confirmed long-lasting limited growth, and despite high year-to-year ring width variability, synchronous growth at vertical study sites dominantly controlled by climate. The positive tree ring growth response to summer temperature was significant for most of the 20th century but dramatically changed in recent decades, when unusually warm summers were reported. There was no, or even a negative growth rate correlation with precipitation, which indicates a sufficient water supply at the study sites. Elevated temperature in this region with a continental climate might turn the study localities to water-limited areas with many negative consequences on tree growth and related ecosystem services.

**Keywords:** boreal forests; Putorana plateau; climate change; tree rings; divergent growth; growth periodicity and synchronicity

## 1. Introduction

Boreal forests are widely spread across northern parts of Eurasia and America and form approximately 30% of the total world forest area. The amount of carbon stored in biomass and soil plays a crucial role in terrestrial C sequestration and land–atmosphere energy exchange [1]. Elevated $CO_2$ concentrations stimulate temperature increases worldwide. Warming is most pronounced at high latitudes where temperatures over the last 30 years have risen by 0.6 °C per decade, which is twice as fast as the global average [2,3]. On the one hand, evident warming occurs at sites where continental climate prevails and oceanic influence is minimal, e.g., the continental part of Asia, including Siberia [4,5]. On the other hand, regional trends strongly vary due to local orography or large water bodies that affect climate dynamics [6].

High-latitude forests are commonly limited by cold temperatures, short growing seasons and a short supply of nutrients [7,8]. Increased temperature might result in increased boreal forest productivity and spatial expansion. This trend has been proven

in many studies in circumpolar forests in Europe, Asia and North America [3,9–11]. The response of tree growth and forest productivity to rising temperatures shows high spatial and temporal variability that is consistent with the observed rates of recent [12] and past [13] temperature changes. On the other hand, some boreal forest ecosystems, particularly in continental interiors, lost their positive response to rising temperatures during the late-20th century [14] or exhibited no trend or a significant downwards trend, particularly after 1990 [15]. While tree growth and forest expansion are mostly controlled by summer temperature and precipitation, several studies point to the role of winter conditions in polar regions as they affect soil hydrological and thermal regimes [13,16]. Additionally, other factors (e.g., slope aspect, geomorphic processes, snow cover, albedo, soil temperatures, carbon and nutrient cycling) can determine the rates of these changes [17].

Tree ring data are widely used for the reconstruction of past climate variability. Especially at the forest limit, variation in tree ring width (TRW) series can be attributed to climatic variation, as the influence (competition) of neighboring trees is limited in wide inter-tree spacing [18]. Annual wood formation is a proxy for tree growth drivers and assists in predicting the reaction of forests to anticipated climate change [19]. High-frequency variability in tree rings is linked with year-to-year climate variability. Low-frequency changes are age linked, and detrending is needed to remove non-stationery processes (radial growth of young trees is faster than that of mature trees) from tree chronologies [18,20]. Traditionally, tree ring-based studies correlate long-term relationships between average tree growth and climate parameters but pay less attention to other less evident growth-limiting factors. Tree growth responses in so-called "event" and "pointer"year analyses refer to remarkable growth increases or reductions and can be used as an ecological indicator of extreme events at the tree or stand level [21].

Tree ring studies have identified similarly increasing tree growth responses to temperature. Common climate trends are assumed to homogenize growth across spatial scales [22]. Increasing evidence has revealed the opposite response of TRW to increasing temperatures. Especially at many boreal sites, the response of trees to temperature changed after the mid-20th century. Correlations with temperature weakened or, in some cases, changed direction (from positive to negative) [14]. A number of causes have been proposed for the divergence of tree growth and temperature, including warming, water deficit and pollution stress [23–25]. There is no consensus on the relative importance of each factor, and the spatial extent of the growth declines has not been fully described. Another source of bias in the detection of natural ecosystem growth fluctuations is the impact of anthropogenically forced environmental changes [15].

We assessed tree growth in the extremely remote and isolated Putorana Mountains, northwest Siberia. We studied tree growth–climate relationships in Siberian spruce (*Picea obovata* Ledeb) within a population located at its natural limit distribution, both altitudinal and latitudinal, where increased sensitivity is expected. Our study site on the western side of the Putorana Mts. was located on the border between oceanic and continental polar climates, featuring a frequent occurrence of extreme weather. Rough topography, steep slopes, fjord-like valleys and large lakes increase meso- and microclimate variability, which is well known for its strong temperature inversions [26] when air temperature increases with altitude. The aim of our study was to estimate the following: (1) growth trends of Siberian spruce at the edge of its natural distribution; (2) altitudinal and site-specific differences in growth trends and climate response; and (3) growth periodicity along different scales for separation of (a) year-to-year weather, (b) climate change and (c) disturbance-induced variability. Based on a literature review, we hypothesized that the central-western Putorana Mts experienced warming accompanied by increased growth of Siberian spruce. On the other hand, we expected that very specific site conditions in the Putorana Mts might split tree growth–climate relations, as the *Picea* genus is known for its divergent response to the changing climate.

## 2. Materials and Methods

### 2.1. Study Area

The study was conducted in the Putorana Mts (Plateau) (Figure 1), northwestern Siberia, which lies on the boundary between the Atlantic and continental Siberian climate provinces [26]. The mean annual temperature and total precipitation are –10.1 °C and 4567 mm, respectively. The cold subarctic climate sustains a continuous permafrost table at 1–3 m depth. The Putorana Plateau is the largest monolithic mountain range in the Russian polar region and one of the most difficult to access regions, making it a poorly studied area. Multiple uplifts on the basaltic Putorana Plateau have generated flat-topped landscapes averaging 900–1200 m in elevation and deep valleys. Forest spreads up to 500 m a.s.l where the upper boundary meets the tundra. Forest vegetation cover is dominated by larches (*Larix sibirica* Ledeb, *Larix gmelinii* Rupr.) and Siberian spruce (*Picea obovata* Ledeb.) interspersed by birch (*Betula* sp.). Soil cover forms basalt-derived Eutric Cambisols (according to World Reference Base) with high stony content [27].

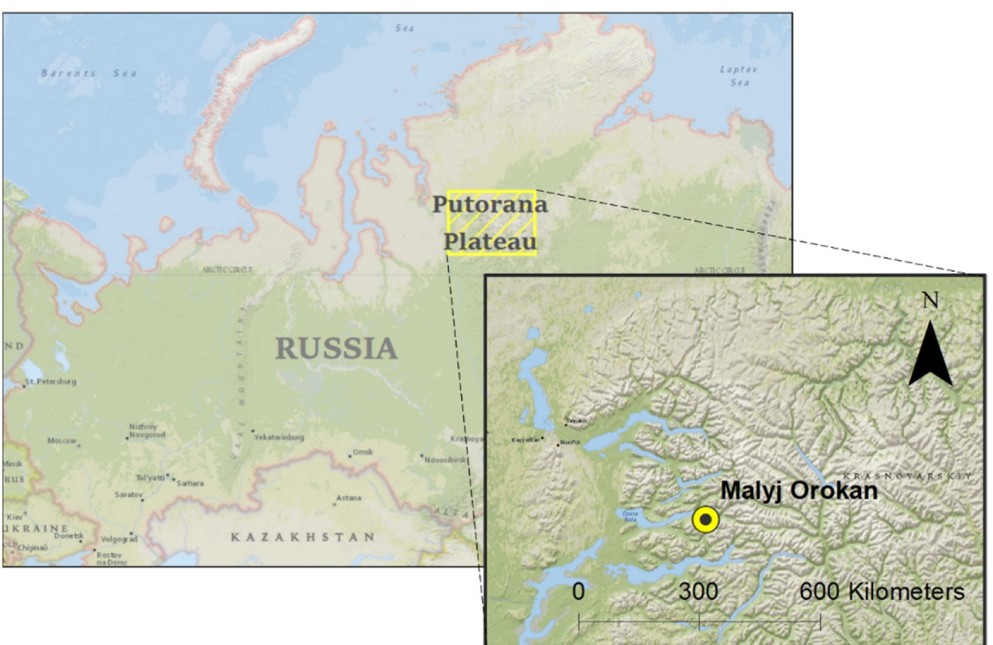

**Figure 1.** Location of the study site in the Putorana Mts, near Keta Lake, Malyj Orokan River valley.

The study locality was established at the western border of the Putorana Mts, approximately 250 km inside the mountains. Three study sites were established along southwest-oriented slope above the deep valley of the Malyj Orokan River (68.68–68.76° N, 91.49–91.55° E) at altitudes of 100, 200 and 350 m a.s.l. The slope angle ranged from 2 to 24°. At each altitude, 5 circle plots covering 500 m² were established. The occurrence of a minimum 5 mature Siberian spruces controlled the exact location of the sampling plots. More than 150 characteristics describing the site, terrain, soil, ground and tree vegetation were assessed in the field (details in [28]).

### 2.2. Dendrochronological Data

Sampled trees were cored with a 5 mm drill 1.3 m above the ground. Cores were tightly wrapped in paper and transported to the laboratory for sanding according to standard procedures. Ring boundaries were identified with binocular analysis and then measured with 0.01 mm accuracy with a WinDendro Image Analysis System [29]. A calendar year was assigned to each ring. All series were visually cross-dated in the COFECHA [30] program and corrected if necessary.

### 2.3. Climatic Data

Monthly air temperature and precipitation data were retrieved from the CRU TS Monthly High-Resolution Gridded Multivariate Climate Dataset 0.5 × 0.5 cell, Central Latitude 68.75 degree, Central Longitude 89.75 degree, version 4.05, period 1901–2020 [31].

### 2.4. Statistical Analysis

Trends in tree ring width series (TRW) were calculated from chronologies after power transformation and detrending. Negative exponential curves were used to remove age-related growth trends in tree ring series (ring width index, RWI). The mean inter-series correlations (Rbar) and the expressed population signal (EPS) were used to check the quality of the chronologies [19]. We computed EPS from the period 1600–2019 in a 10-year window using rwi.stats.running function in dplr R package, version 4.0.3. An EPS of 0.85 was chosen as an appropriate criterion to ensure the reliability of the chronologies.

For visualization of a tree ring chronology in the frequency domain, continuous wavelet transformation [32] was performed, and for plotting, the wavelet.plot function from dplR R package, version 4.0.3 was applied.

For wavelet transformation and plotting the function morlet.plot from the R package dplr was used. A total of 0.1 suboctaves per octave were assessed, and the significance level was set to 0.99.

Wavelets were also used for decomposition of a TRW time series into specific frequency components using the multiresolution analysis (mra) of time series function in the waveslim R package, version 1.8.2.

Specific frequency components were used for event or pointer year identification. The threshold values to consider a relative growth change (in percent) as a negative/positive event year were set to 40 and 60, respectively, in point.res package, function pointer.rgc. The minimum ratio of trees with extreme values was set to 1.645 (default value in pointer.norm function).

Only positive and negative extremes were used for visualization. Event years were identified for a single tree, and pointer years were identified for multiple plots at the study sites.

For the detection of ecological gradients in tree ring series that express differing trends, principal gradient analysis (PCGA) was used. The analyses were performed and plotted by the R package dendroIAB version 0.2. The identification of differences in TRW synchrony was calculated by the Wilcoxon rank sum test (wilcox.test function) in the dendroIAB R package. Time intervals were set to 50 years, as the temperature variability in boreal regions showed a similar pattern [13].

Common (synchronous) patterns in tree ring width variation can be disentangled and interpreted by grouping chronologies into potentially homogenous subsets. Using DendroSync [33], we assessed to what extent temporal responses are spatially structured by partitioning the variability associated with the time effect at altitudinal study sites (intra-) and between study sites (intergroup). The "âC" values explain synchrony between RWI series calculated by covariance–variance mix model. For trends detection and plotting the results, we used the synctrend and synctrend.plot function in the dendroSync R package, version 0.1.3.

Trends in climatic series were tested by the Mann–Kendall trend test in the kendall R package, version 2.2.

Running correlations between climatic variables and detrended chronologies RWI were analyzed by the dcc function from the bootres R package, version 1.2.4. To identify temporal changes in large detail, we set the window size to 18 years (the most detailed resolution in sw) and window offset to 2 years.

## 3. Results

### 3.1. Tree Ring Width

The average tree ring widths (TRW) showed relatively similar growth over decades at all three altitudinal sites. Mean TRW (standard deviation in brackets) for the down (KD, 100 m a.s.l.), middle (KM, 200 m a.s.l.) and upper (KU, 350 m a.s.l.) study localities were 0.36 (±0.16), 0.31 (±0.14), and 0.28 (±0.10) mm, respectively.

The inter-annual TRW variation among study plots shows a time series plot of running standard deviations (Figure 2). From the 1550s until the 1850s the TRW values were very similar. Later, notable deviation has occurred at the KD site. The KM and KH shows relatively stable growth during almost a whole observed period.

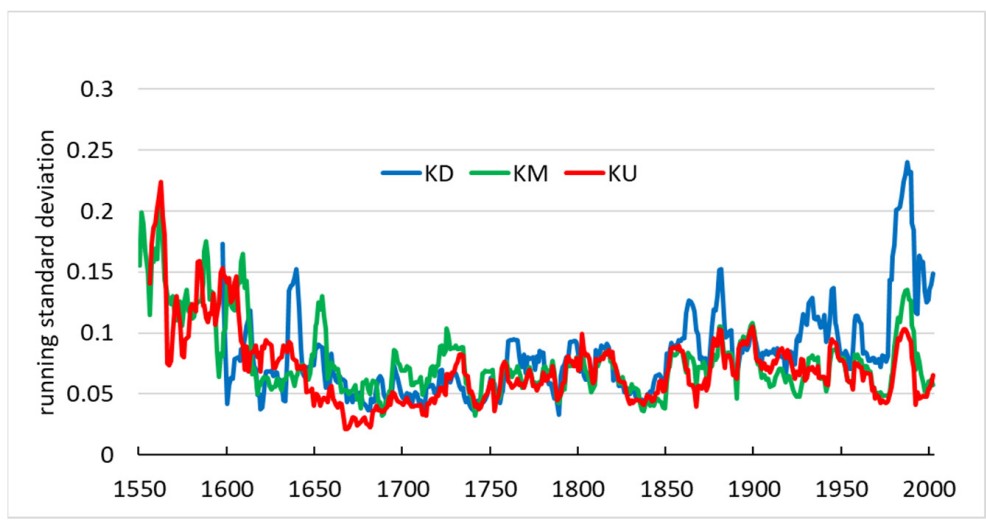

**Figure 2.** TRW running standard deviation with 10-year window at the altitudinal study sites.

The number of sampled trees and the distribution of individual tree ages at each altitudinal study site showed slightly different patterns (Figure 3(Ab,Bb,Cb)). At the lowest study site (KD), almost 50% of trees were established between 1810 and 1840. Trees older than 210 years were evenly distributed up to the oldest category. At the KM site, nearly 30% of trees were 250 years old, which means that they were established at approximately 1770. Both younger and older trees were proportionally distributed between the youngest and the oldest trees. At the highest, KU site, nearly 40% of sampled trees were 260–280 years old, established in 1740–1760. Additionally, other trees were evenly distributed as at the lower study sites.

The tree age structure at the study sites might notably influence the interpretation of growth rates. The growth rate with influence of increasing age on tree ring formation removed (ring width index, RWI) is presented in Figure 3(Aa,Ba,Ca). The grey line is detrended (master) chronology representing the values of all trees at each study site. Blue dots are a measure of the representativeness of the derived master chronology, expressed population signal (EPS). High values (close to 1) indicate increasing synchronicity in TRW data. Red line is the 20-year spline of master chronology.

Synchronicity in the tree ring growth rate is a prerequisite for the representativeness of RWI chronologies. Synchrony at each vertical study site was classified using EPS. High (close to 1) EPS signal values mean that at each altitudinal study site, a large number of sampled trees reacted synchronously, and TRW data should be averaged to construct a so-called master chronology. Figure 3(Aa,Ba,Ca) show how tree ring chronologies contain a common signal over different time periods. A synchronous response between sites is evident from the 1880s.

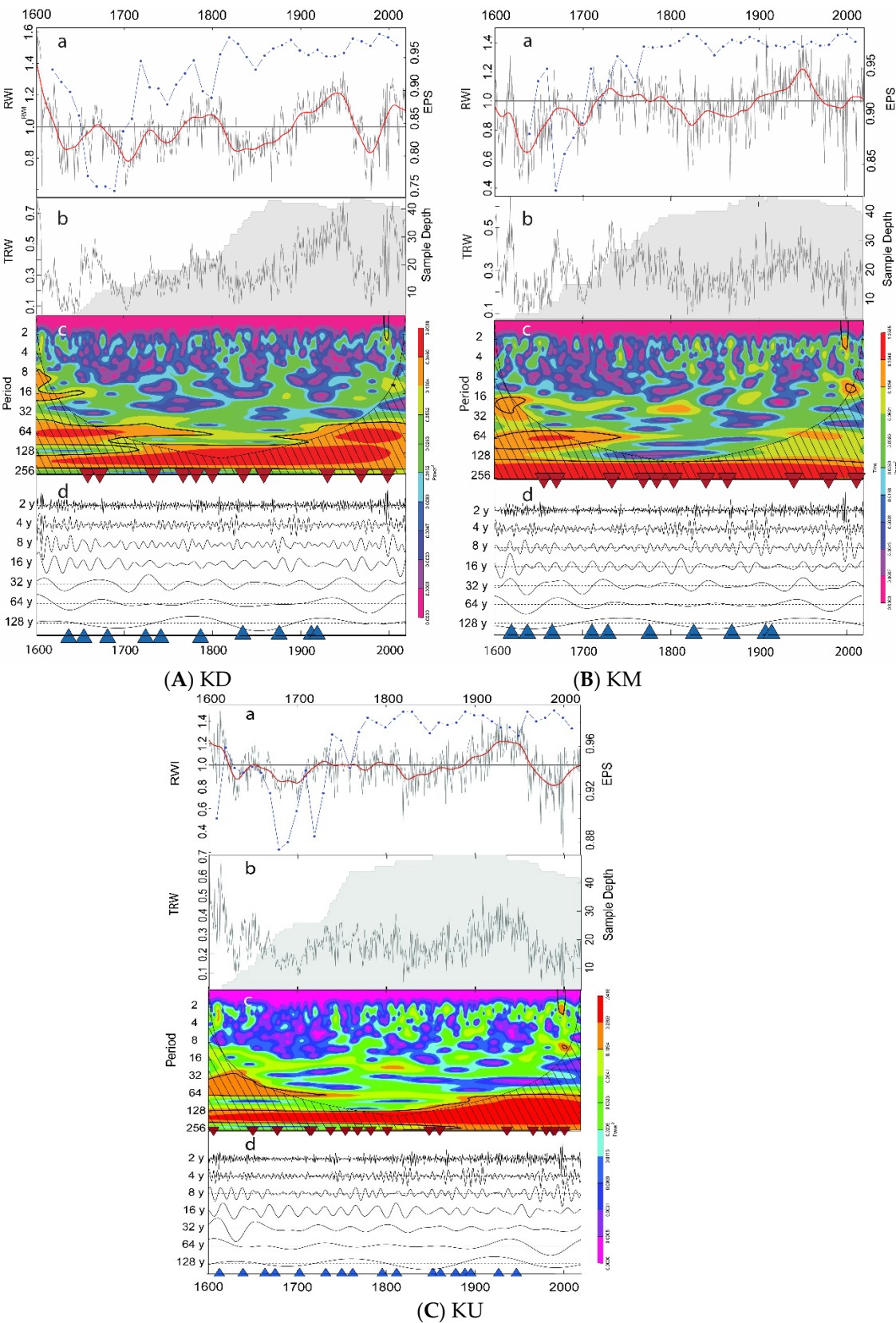

**Figure 3.** (**A**–**C**) The three study sites—KD, KM and KU: (**a**) the detrended ring width (RWI, grey line), 20-year spline (red line) and EPS (blue dots); (**b**) the raw TRW and sample depth; (**c**) the Morlet power spectrum; and (**d**) the multiresolution decomposition of TRW and pointer years (red triangle—negative year, blue triangle—positive year).

Wavelet analysis showed periodicity in the tree ring growth rate at three study sites on a multiresolution scale ranging from 2 to 248 years (left y-axis) for the 1600–2019 period (x-axis). Significance of the periodicity explained power spectrum values ranged from low (purple) to high (red) (right y-axis). The most significant periodicity is outlined by the black line. Dashed lines mark edge effects where periodicity is irrelevant for the 1700–2019 period (x axis). Significance is reported by color bars. Only centennial periodicity was significant. Less significant periodicity in tree ring variation is detectable on different time scales, ranging from years up to decades or centuries. High-frequency signals reflect annual weather variability, while lower-frequency signals reflect long-lasting processes, such as stand development or changing climate. Visual classification can identify similarities or differences mostly in low-frequency space. On the centennial scale, a notable TRW increase occurred in the mid-18th century and a decline in the mid-19th century. The identification of high- and low-frequency signals in TRW records was crucial for the identification of year-to-year climate variability, long-lasting climate variability and disturbance events. Pointer years occurred synchronously at the KD and KM sites. KU experienced more pointer years. Across all study sites strong events probably occurred in 1874, 1875, 1908, 1982, 1984, 1986, 1989 and 2000. The variation in the TRW signal on different temporal scales was classified by multiresolution decomposition of TRW. Figure 3A–C/d show TRW variation on 2-, 4-, 16-, 32-, 64- and 128-year scales for all three altitudinal study sites.

Despite high variability in RWI during the observed period (1700–2018), synchronous declines at approximately 1800 and 1950 at all study sites were evident, and the most apparent decline occurred for KD. A similar drop occurred in 1950 and lasted until the 1970s. The last decline occurred in 2000 and lasts to present.

For classification of synchronicity, or identification of hidden divergence in tree ring growth rate between study sites along altitudinal gradient, we applied principal component gradient analysis (PCGA). Figure 4 shows the PCGA analysis performed in 50-year step for the period 1700–2019. The PCGA showed only weak differences among study sites at different altitudes. Heterogeneous micro-topographical (orientation, slope), soil (stoniness, physical and chemical properties) and stand characteristics (biomass, stand density) did not play any role in splitting individual RWIs into homogenous groups.

The statistical significance of differences in synchrony between study sites was classified with the Wilcox rank sum mean test (Table 1). The results confirmed the synchronous tree ring growth rate at KM and KU and slight differences at the KD study site.

Despite differences in absolute TRW values, synchronicity among sites prevailed during 1700–2019.

For temporally more detailed analysis and interpretation of synchrony patterns in tree rings, we used data for the period after 1900 (when meteorological data were available) and applied "DendroSync". The results show to what extent fluctuation in tree ring data (RWI) can be disentangled and grouped into potentially homogenous subsets (Figure 5). Synchrony was classified within groups (RWI on five sampling sites at each elevation zone) and between groups (three elevation zones).

Temporal changes of synchrony (âC) showed an increasing tendency at both altitudinal (within) and whole study (between groups) levels until the 1990s. After this date, synchrony started to decline and a slight divergence between KD and other localities could even be identified.

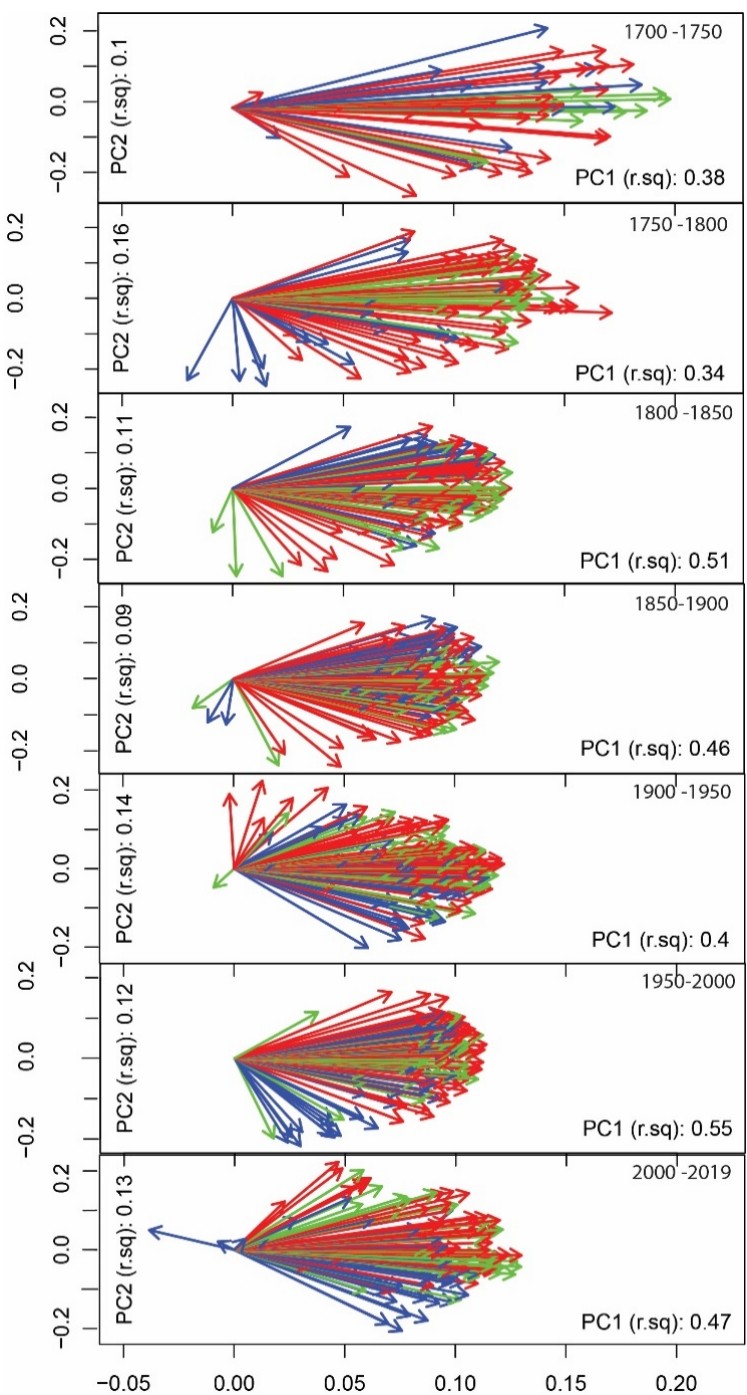

**Figure 4.** PCGA synchronization of the tree ring growth rate during the 1700–2019 period in 50-year steps. Altitudinal study sites are differently colored: KD (blue), KM (green) and KU (red). Each arrow represents individual tree; the arrow direction indicates TRW synchrony, and the length indicates the TRW growth rate. The PC1 (x-axis) and PC2 (y-axis) are shown in each 50-year period figure.

**Table 1.** Difference of synchrony in TRW, PCGA and Wilcox, * = sig > 0.05, NS = non sig.

| Period Site | 1700–1750 | 1751–1800 | 1801–1850 | 1851–1900 | 1901–1950 | 1951–2000 | 2000–2019 |
|---|---|---|---|---|---|---|---|
| KD-KM | NS | * | * | * | NS | NS | * |
| KD-KU | NS | * | * | * | * | * | * |
| KM-KU | NS | NS | NS | NS | NS | NS | NS |

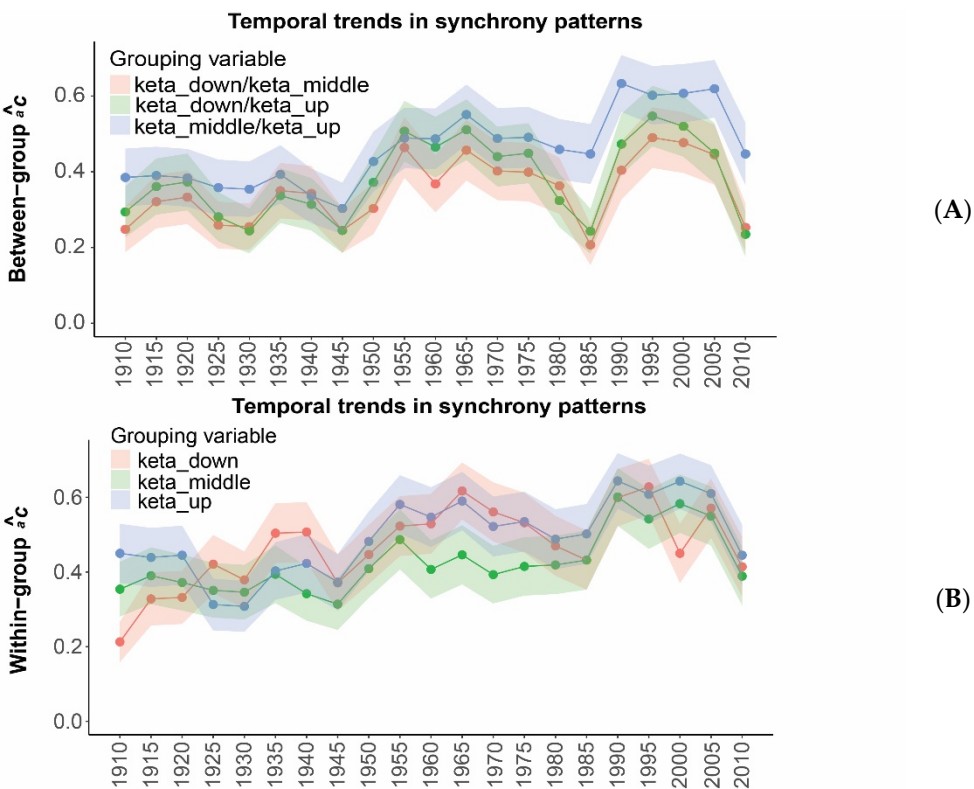

**Figure 5.** Synchrony estimates (âC) for tree ring width at (**A**) within-group and (**B**) between-group levels are calculated for the best model for 30-year moving intervals lagged by 5 years over the period 1900–2019. The x-axis shows the central year of the moving time interval. Grouping of chronologies is based on the position of study plots along a vertical gradient (KD—blue, KM—green and KU—red). Shadows are standard errors (SE).

*3.2. Climate*

Mean monthly air temperature and monthly precipitation sums were retrieved from gridded climate data (CRU TS database), Figure 6A,B for the period 1901–2020. Both air temperature and precipitation showed significantly increasing trends ($p < 0.001$, Mann–Kendall trend test).

Differences between 1951–1980 and 1981–2010 (according to the World Meteorological Organization (WMO) Standard years) are displayed in different colors (Figure 7). Air temperature increase is evident throughout the year. The difference was larger in winter months (NDJFMA) ranging from +0.5 to 2.2 °C. During summer (JJAS), the temperature increase ranged between 0.1 and 0.7 °C. Statistically significant differences are denoted by asterixis (*t* test, $p < 0.05$). In the average period, 1981–2010 was warmer by 1.1 °C than 1950–1980. Summer temperatures are understood to be critical for tree growth in polar regions. June and July temperatures are often used to identify climate change in impact studies. The decadal annual mean increased from −11 to −7.8 °C between 1901–1910 and 2011–2020.

Annual precipitation sums between two standard WMO periods did not differ notably (486 vs. 505 mm). Nevertheless, winter months experienced higher totals, on average +10%. The largest difference occurred in February (increased by 29%). The most evident precipitation decline occurred in September (minus 24%). In warmer summer months (JJA) precipitation totals remained almost stable (±5% in individual months).

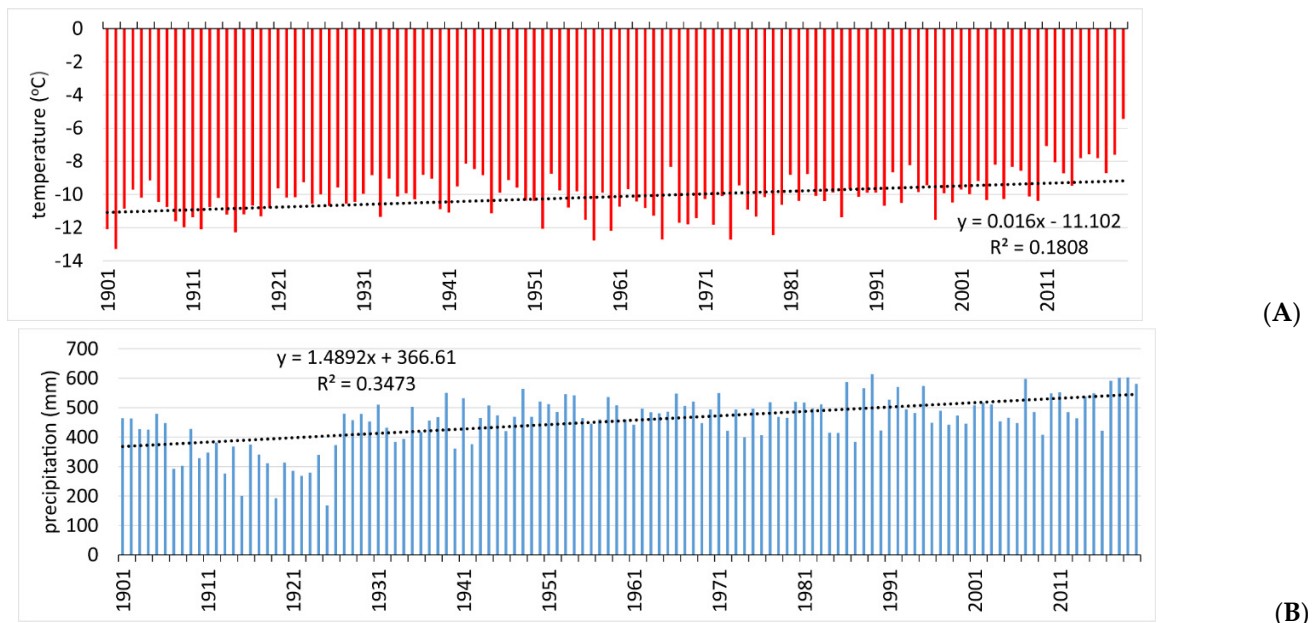

(A)

(B)

**Figure 6.** (**A**)Annual mean temperature, (**B**) precipitation totals and linear trends. Data derived from gridded CRU TS database v. 4.

(**A**)  (**B**)

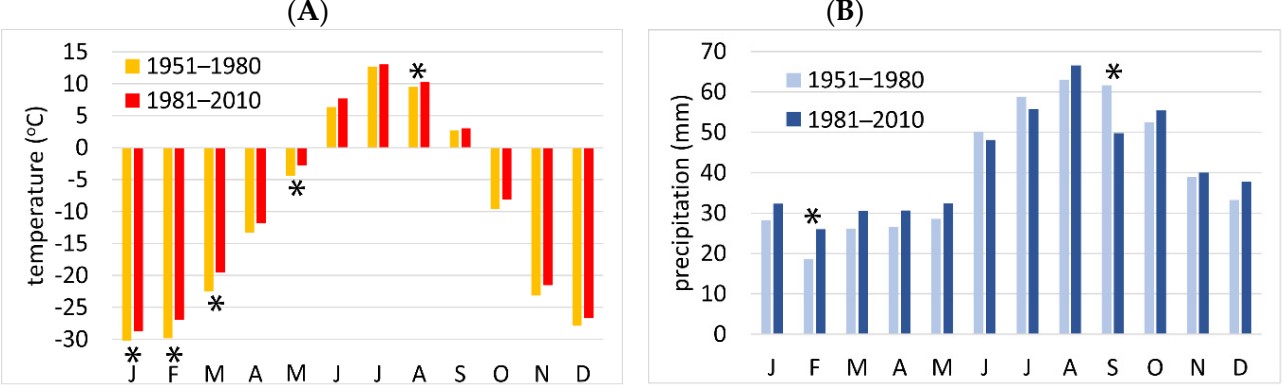

**Figure 7.** (**A**) Average monthly temperature and (**B**) precipitation totals for WMO standard periods 1951–1980 (light) and 1981–2010 (dark). Statistically significant difference ($p < 0.05$) is denoted by asterix (*).

### 3.3. Tree Rings and Climate

Running correlation was used to identify the relationship between the tree ring growth rate and climate parameters. As the climate data only go back to 1900 and the study sites showed synchrony in growth rate between study sites (PCGA and DendroSync analysis), the running correlation analysis yielded very similar results. In Figure 8, running correlations for the KM study site are shown for previous autumn (SO), current summer (JJA) and current spring (AM) temperatures and precipitation totals.

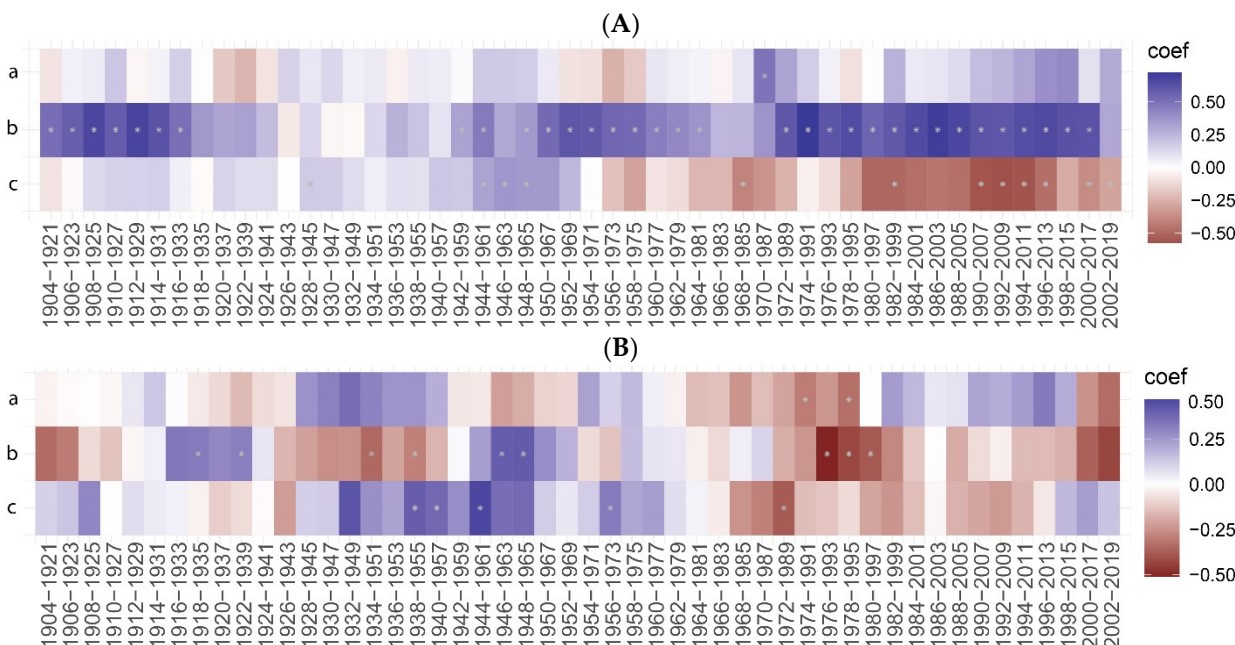

**Figure 8.** Running correlation analysis with an 18-year window and 2-year step shows (**a**) previous autumn, (**b**) current summer and (**c**) current spring intervals between the tree ring growth rate and (**A**) temperature and (**B**) precipitation. Correlation is depicted by colors. * = significant *p* < 0.05.

From 1900, tree ring growth was positively correlated with temperature. Significant correlation prevailed. A relatively strong correlation (r ≥ 0.5) started in the mid-1970s and lasted until the mid-2010s. Similar correlation was identified in 1910s–1930s. Surprisingly, a similar, but negative correlation showed increasing spring temperatures, which culminated in 1990–2010 and to some extent continue to present days. Tree ring growth has rarely been correlated with current year precipitation. Previous autumn and mostly current spring precipitation in the past (1940s–1950s) used to be more relevant than during the last 50 years.

Despite warm and sufficiently wet conditions in the last 20 years, tree ring growth declined, as shown by TRW and RWI. The running correlation revealed that the correlation with climate parameters is also declining.

## 4. Discussion

### 4.1. Tree Ring Width (TRW)

Tree ring width as an indicator of tree growth confirmed unusually limited growing conditions at the study sites in Keta Lake region. We averaged TRW even though tree ring growth is nonstationary process and to average TRW together is not always recommended. Our approach was supported by long-lasting temporal stability of inter-annual TRW variation and relatively uniform [27] stem sizes at the study plots. Centuries of Siberian spruce TRW averages ranged between 0.36 and 0.28 mm along the altitudinal transect from the valley bottom close to the tree line. Even during the most intensive growth in the early 20th and 21st centuries, the annual TRW hardly reached 0.8 mm. This value for Siberian spruce (0.8 mm) was reported [15] as an average TRW for the study belt in boreal forest >65° N spreading from Fennoscandia up to Eastern Asia and [11] from the taiga–tundra ecotone of the Urals. According to [13], the mean tree ring width for *Pinus sylvestris* along the Arctic margin in Scandinavia and Kola Peninsula (>67° N) ranged between 0.66 and 1.0 mm. As the growth intensity depends on tree age, it should be mentioned, that tree series lengths differed notably among the mentioned localities. Much younger stands were sampled in the Urals (average 80 years), and in the Scandinavian-Kola study, the tree age ranged between 166 and 272 years.

The average age of the sampled trees in our study at the Keta Lake region was 281 years with a maximum 461 years. We found increasing age with altitude, but age span and age variance showed opposite directions. This tendency differed from studies reported in Northern Putorana [34] and in the Urals [9] where young trees dominated in forests near tree lines. As these authors concentrated on tree line migration and succession in tundra, they probably paid more attention to younger trees.

The tree age distribution along the altitudinal transect revealed some step-like periods. We understand the presence of such homogenous age groups as an indicator of past forest external impact, which first disturbed and consequently initiated stand regrowth. The possible impact of such events was confirmed on surrounding trees. Negative pointer years coincided with the beginning, and positive pointer years occurred during the mentioned periods, when trees profited from changed conditions, as discussed later.

*4.2. Climate*

Despite year-to-year variation, the general warming trend in meteorological data since 1900 is clear and supported by surrounding field measurements [34,35] Observational meteorological data for the studied western and central parts of the Putorana Mts do not exist. The nearest station with a decades-long history is in Dudinka (ca. 300 km west direction). For our study, we used gridded climate data from the CRU TS v. 4.05 database. Temperature and precipitation trends derived from the CRU database confirmed general increasing temperature and precipitation totals since 1900. Grids $0.5 \times 0.5$ degree are relatively large. To what extent the specific environmental conditions of the Putorana Mts [26] are reflected in gridded climate databases is unclear. We speculate that plateau-like mountain topography is better aligned with climate models compared to mountains featuring jagged topography.

Most published papers from European and northwest Siberian boreal forests identify nonsignificant temperature increases [15,34,35]. A climadiagram constructed from the Urals by [11] with data from the Salekhard climate station ($66.32°$ N, $66°$ E, period 1892–2015) was similar to our data shown in Figure 6 despite the notable distance between sites. In particular, both growth and dormant period temperatures increased by 1.7 °C since 1900. Data from the Urals showed significant warming and moistening over the last 135 years. The precipitation increase in more continental Putorana Mts was notably lower, e.g., May–August precipitation in the Urals increased from 175 to 224 mm and from 135 to 238 mm in September–April. Contrary to a report by [34] that annual precipitation totals increased, especially during winter months (ONDJFMA), while summer (JJ) totals remained relatively stable, we found a decline in summer months.

A comparison among average decadal temperatures confirmed the increasing pace of warming in the Keta Lake region. Summer temperatures (JJ) increased by 3.0 °C from 8.7 (1900–1910) to 11.7 °C (2010–2020). The more northern Taymir [36] reported a temperature increase even by 5 °C in chironomid-based study. The decadal winter temperature (1901–1910) in our study region increased by 3.8 °C when compared with 2011–2020. This corresponds to the fact, that the most pronounced trend of modern warming in the continental part of Asia, including Siberia, is observed during the cold season (e.g., [4]).

Increased winter temperature and less precipitation are reported to have a devastating impact on boreal forests. Repeated freeze-thaw effects under elevated winter temperatures with insufficient snowpack cause fine root damage [37,38]. This happens mostly in moist boreal forests, different from those in the Keta Lake region. We found south-oriented stony slopes with shallow soils predisposed to water deficit. The annual precipitation is approximately 400 mm, which is understood to be enough to cover Siberian spruce needs [39]. We question this statement under episodes with strong inversions when the temperature gradient exceeds 6 °C per 100 m as reported by [26].

### 4.3. Climate Tree Growth

Our data confirmed a positive relation between temperature and tree growth, consistent with the well-known temperature constraints of boreal forests. Warming extends the growing season and increases growth rates while reducing potential cold-temperature injuries. Moderate increases in temperature (1–2 °C) are understood to have positive effect on boreal tree growth [40]. During the last 120 years, the temperature in the study region smoothly increased by 1.6 °C, but the Siberian spruce growth response did not fully follow the warming trend. Both raw TRW and detrended WRI showed increasing growth until the 1950s, and then a steep decline occurs. The weak temperature–growth running correlation in the 1930s–1960s (Figure 7) relatively well covered and explained the cool period from 1940 to early 1980s reported by [13]. During the warmer period of late 1970–2010 (identified by the same authors) in 1985–2006, tree ring size increased, and the correlation temperature–growth rate was rather strong. The positive effect of warming between the 1970s and 1990s and precipitation increases since the 1950s on Siberian spruce and Scots pine growth was identified [35] in northern taiga and forest–tundra zones in Komi (>65° N, NW Russia). In the last very warm decade, both tree ring size and correlation with temperature dramatically declined.

A relatively weak growth response to increasing temperature was reported from large-scale studies (e.g., [41]) as well as from local studies (e.g., [10]) in Finland, where spruce was even less sensitive than pine or birch. A general decline of boreal forest temperature response from the late 20th century has been reported [1]. The authors identified an increasing dependence on soil moisture as a factor counterbalancing the potential positive effect of warming on boreal forests. Annual totals above 400 mm are understood to be sufficient for Siberian spruce growth in cold boreal conditions [39]. Precipitation in the Keta region is slightly above this level. Increasing temperature might reduce available sources and cause discrepancies in tree transpiration needs. A decline in water-stressed Siberian spruce growth on steep stony slopes and convex terrain elements across Central Siberia has been reported [42].

Surprisingly, higher precipitation in winter months had no positive effect on tree ring growth in our study. It is well known that precipitation from the previous autumn to the current spring provides the necessary moistening of the soil at the beginning of the growing season, which is beneficial and highly important for tree growth [43]. In turn, high summer temperatures in conjunction with low precipitation can affect the production of sugars and be considered a manifestation of drought stress [44]. We believe that precipitation is not a limiting factor for Siberian spruce growth at the Keta Lake study site. A weak, or even negative response of tree radial growth to precipitation amount during the period after 1900, when meteorological data were available, indicated a sufficient amount of precipitation. On the other hand, in the cool period, around 1950, tree ring growth was positively correlated with precipitation (as shown in Figure 8B). Weak relationships with precipitation for Siberian spruce have also been reported [11]. A temperature–growth simulation by [40] indicated that additional warming of 3–4 °C may lead to substantial growth declines in continental boreal stands.

Tree ring widths in the Keta Lake region showed two exceptional peaks, both well identified at the KD study site, where all five study plots were located at the wide bottom of the Malyj Orokan River on very gentle slopes. Large tree ring increase started in 1870s. Despite big difference in absolute tree ring widths the growth trajectories identified by the growth trend among all study sites remained similar. We understand this response as the prevailing impact of climate, which dominantly controls tree ring growth. Very likely warming at the end of the Little Ice Age (1870–1939), as reported by [13], released water from snow and frozen soil and led to huge transport of soil and nutrients downwards. The synchronous decline in tree ring widths at all study sites after 1940 correlated with the cool period. Another peak in the tree ring growth rate occurred in approximately 2000 and was synchronous with the elevated temperature in 1985–2006 reported by [13]. Again, the most evident TRW increase was at the lower (KD) site. We cannot identify whether it was

caused by another repeated flood fertilizing the valley bottom, or just a legacy of former floods. Almost flat terrain and a relatively large distance from the steep upper mountain valley likely limited the destructive power of floods, and disturbance signs in tree ring chronologies were rather sparse. Based on the $Na^+$ concentration in soils at these sites, the authors of [28] reported periodic external sources of nutrients. Large stony streams along rivers and the widely spread rough microtopography on the valley bottom support this interpretation.

We found rather weak agreement between extreme climate conditions and pointer years in tree rings chronologies. One of the reasons might be insufficient representativeness of gridded climate data for specific study site conditions, as variation of warm/cold and dry/wet condition extremes and periodicities detected in tree rings series are often in good agreement and confirmed reliability of detected climatic signals [43]. Another explanation is the occurrence of large water bodies (Keta Lake and many others in its vicinity), which buffer magnitude in the air temperature fluctuation and contribute to moisture content in the air through evaporation [6].

We hypothesized divergent growth of Siberian spruce as a consequence of contrasting site conditions along altitudinal transect in the Keta Lake region. The analysis (PCGA) did not confirm this expectation, which we understand as solely climate-driven growth conditions in the study region. Extreme weather changes probably limit tree ring growth permanently, which masks processes that are otherwise more visible. Some signs of starting divergence were found at the KD site, which might respond to both increasing temperature and nutrient enrichment under irregular floods. Further increasing the temperature might substantially influence the water regime and strongly affect Siberian spruce growth. The present growth decline and synchronous responses are indicative of processes that might reduce the ecosystem services of Siberian spruce for climate change mitigation [45].

**Author Contributions:** Conceptualization, P.F. and V.P.; methodology, P.F. and P.F.J.; formal analysis, P.F. and V.P.; investigation, P.F., J.M., V.P., E.G. and M.H.; data curation, P.F. and P.F.J.; writing—original draft preparation, P.F. and P.F.J.; writing—review and editing: P.F., P.F.J. and V.P.; supervision, V.P.; project administration, V.P.; funding acquisition, V.P. All authors have read and agreed to the published version of the manuscript.

**Funding:** This research was funded by the Slovak Research and Development Agency of the Slovak Republic, Project No. APVV-17-0676 and APVV-17-0644.

**Data Availability Statement:** Data are available from corresponding author upon reasonable request.

**Acknowledgments:** The organizational, logistical support and help from Konstantin Prosekin and Alexander Matasov are gratefully acknowledged.

**Conflicts of Interest:** The authors declare no conflict of interest. The funders had no role in the design of the study; in the collection, analyses or interpretation of data; in the writing of the manuscript, or in the decision to publish the results.

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
