# Peer review of "Declining Growth Response of Siberian Spruce to Climate Variability on the Taiga–Tundra Border in the Putorana Mountains (Northwest Siberia)"

_forests, doi:10.3390/f13010131_

Round 1

Reviewer 1 Report

The manuscript “Declining growth response of Siberian spruce to climate variability on the taiga-tundra border in the Putorana Mts (north-west Siberia)” by Peter Fleischer, Viliam Pichler, Ján Merganič, Erika Gömöryová, Marián Homolák and Peter Fleischer jr. describes research of the spruce radial growth’s temporal stability near its northern boundary under impact of climate warming. The topic is relevant and proposed statistical approaches well encompass phenomenon of the divergence in tree growth–climate relationship in the framework of selected study area and tree species.

Title reflects the contents of study adequately. Abstract could be more precisely formulated; background information presented there is somewhat vague and described in excessive length. Introduction represents state of art and supported suitably by citation of modern studies. However, it is somewhat superfluous and should be made more short and precise. Materials and Methods are adequately described, selection of study object (region, sampling sites, species) is justified, but authors need to pay more attention to citation of proper data sources, R packages, etc. In Discussion, much more citations should be added for statements not obvious from presented data. Also, in sub-section 4.2, most of the first paragraph (L507-517) belongs to Materials and Methods, the next two paragraphs (L519-535) are typical Results. Reference list is not formatted to the journal style, please re-format.

Figures and tables demonstrate important points of the study. However, authors should pay more attention to figure numbering (there is a mistake of two Fig. 1), sizes, and layout on the page. All panels of one figure should be presented on the same page, and caption should be put under it (unlike Fig. 2 on PP. 6-7, where caption is also poorly written). Table1 1 and 2 are so similar and simple that I recommend combining them in one table. Quality of English translation is passable (text is comprehensible in general), but manuscript would still benefit from editing by native English-speaking colleague of by English Editing Services.

The text in general contains a lot of repetition of the same word within one or consequent sentences. E.g., in abstract, the word “forest” occurs three times in two sentences: “Response of boreal forests to ongoing changes is an urgent topic as these forests cover 16% of global landmass and play important role in the Earth carbon cycle. Boreal forest growth is generally limited by low temperatures, so increased temperature is supposed to enhance biomass production and carbon sequestration.” It is recommended to rephrase text where possible to avoid such repetitions.

Minor comments:

L36. Delete comma.

L49. When cite several consequent references, write [9-11] rather than [9,10,11].

L56-57. Authors write that “many studies suggest the role of winter conditions in polar regions as they affect soil hydrological and thermal regimes” but cite only one reference [16], and it is not even a review. Pleas add more citations or rephrase.

L67-70. Add citations.

L73-75. “Tree ring studies have identified similar tree response to temperature.” Please rephrase or elaborate, similar to what? Geographical extent of this “similar response” is not mentioned too. The same in the next sentence: opposite to what? In current form these sentences are confusing.

L79. Better to replace the second occurrence of “temperature” with “warming”, or rephrase otherwise.

L80-81. In phrase “nor spatial extent of the growth declines have not been fully described” delete “not”, it is better to avoid double negative.

L91. Not all readers know meaning of term “temperature inversion”, please add brief description.

L98. Write “Picea” (Latin name) in italics.

L102-103. Words “western Siberia” should have commas at both sides.

L108. Write “Plateau” from capital letter.

L112. Write “Betula” (Latin name) in italics.

L112-113. Add source (citation of reference or website) for World Reference Base.

L116,120,482,586,611. Write words “Lake”, “River” from capital letter.

L121. In other places elevations are stated to be 100, 200, and 300 m a.s.l., but here upper site is indicated to be at 350 m a.s.l.

L129. Please add source (citation of reference or website) for WinDendro if possible.

L130. Reference [38] seems to be wrong. If you want to cite work of Grissino-Mayer on COFECHA, it is this: Grissino-Mayer, H. D. (2001). Evaluating crossdating accuracy: a manual and tutorial for the computer program COFECHA. Tree-Ring Res., 57, 205-221.

L133-135. Add source (citation of reference and/or website) and state cover period for CRU TS series.

L152-174. Ensure that all R packages and functions have appropriate citations, like [39] and [40].

L159-161. Please clarify thresholds used to define event (threshold value for extreme) and pointer year (ratio of trees with extreme). You can find good description and discussion of various pointer years’ definitions in this work: Schweingruber, F. H., Eckstein, D., Serre-Bachet, F., & Bräker, O. U. (1990). Identification, presentation and interpretation of event years and pointer years in dendrochronology. Dendrochronologia, 8, 9-38.

L161,602. Correct “point years” as “pointer years”.

L174. Trends should be written in plural form.

L183. Elaborate how TRW was averaged? With arithmetic mean or bi-weight mean?

L187. Add comma after “20th century”.

L198-199. “Next short and steep TRW 198 increased changed direction to decline after 2000.” The sentence is confusing, please rephrase.

L205-206. Write not abbreviated word “significant”, and p<0.05.

L280 and further paragraph. High EPS values can mean high synchronicity OR increase in sample depth OR both. Please demonstrate it more clearly.

L294-296. “Less significant periodicity Tree ring variation is detectable on different time scale, ranging from 295 years up to decades or centuries.” Something is missing in this sentence.

L300. Delete comma.

L303-305. Awkward sentence “Synchronicity across all study sites, thus probably strong events happened in 1874, 1875, 1908, 1982, 1984, 1986, 1989, 2000.”

L315. Why mention DendroSync here? Its usage is described later.

L348-350. Explicitly tell about colors and that arrows represent individual trees.

L352-355. This is basically repetition of figure caption.

L369. The text “(R statistical environment)” belongs to Materials and Methods.

L399,402. “(âC)” – what is this?

L410-411. Write correct period: “2019-2020” seems very wrong.

Figure 5. Align plots together and add axis labels everywhere. Also, it should be annual mean or total, not monthly.

L436. “for identification climate change” – perhaps “for identification of climate change”?

L440-444. It would be nice to estimate significance of differences between periods.

L449. Measurement units should be written only in the respective axis labels, not in the figure caption. Also, change “rain” with “precipitation” in the label, there is certainly snow there.

L463-466. Delete “for 18 year-long intervals”, it is a repetition from previous line. And generally, structure of caption is awkward.

L472. Replace “extend” with “extent”.

L546-549. Add references to both statements.

L571. Correct “moths” as “months”.

L576. “manifestation drought stress” – perhaps “manifestation of drought stress”?

L590. Missing closing bracket.

L595. Correct “increased“ as “increase”.

L617. Correct “might… affects” as “might… affect”.

L629-633. Fill out this statement properly.

Author Response

Dear reviewer,

Thank you for your comment and recommendation for improving our manuscript. Please find our replies enclosed.

In the Revised version 1.1 all the changes required and recommended by the Rev1, Rev2 and the English Editing Service are highlighted with yellow, green, and blue colours respectively. The final text is presented in the Revised version 1.2. Remote access to the office PC due to the restrictions did not allow to use “track changes” mode.

Reviewer 2 Report

The phenomenon of divergence in dendroclimatology has been around for years now, and it remains an important issue to deal with.  Are tree rings changing their long-term relationship with climate because of warming?  If yes, then an underlying principle of dendroclimatology, uniformitarianism, might be ruled invalid.  Also if yes, then divergence itself might be interpreted as yet another line of evidence for warming and its impacts on ecosystem health.  Divergence has mostly manifested itself in boreal chronologies with temperature as the primary limiting climatic factor.  Thus, this ms has the ingredients necessary for making for a publishable article.

A few comments on concepts as well as on the English writing:

It was with anticipation that I got to the end of the ms to see if divergence would be declared an issue at this site or not. Lines 559-560: “In the last very warm decade both tree ring size and correlation with temperature dramatically declined.” This seems to imply yes, divergence is in effect here.  But then, lines 610-612: “We hypothesized divergent growth of Siberian spruce as a consequence of contrast site conditions along altitudinal transect in Keta lake region. The analysis (PCGA) did not confirm this expectation.”  Now I’m not so sure.

Line 126: “Sampled trees were cored with 0.6 mm drill.”  0.6 mm?  Diameter?  Typical increment borers come in two diameters: 4.2 mm or 5.0 mm.  0.6 mm in diameter is almost nothing, hard to believe that’s the correct number.

Line 130, the citation #38 is for a sentence about COFECHA, but the paper #38 is about the ITRDB, not the same thing. I’d recommend one of these two:

Grissino-Mayer, H.D. 2001. Evaluating crossdating accuracy: A manual and tutorial for the computer program COFECHA. Tree-Ring Research, 57(2), 205-221.

Holmes, R.L. 1983. Computer-assisted quality control in tree-ring dating and measurement. Tree-Ring Bulletin, 43, 69-78.

Figure 1: From the Cook and Kairiuskstis book (Cook, E.R., Briffa, K.R., Shiyatov, S., Mazepa, V. 1990. Tree-ring standardization and growth-trend estimation. Pages 104-123 in: E.R. Cook and L.A. Kairiukstis, eds., Methods of Dendrochronology: Applications in the Environmental Sciences. International Institute for Applied Systems Analysis, Kluwer Academic Publishers: Boston, MA.): “Strictly speaking, it is not valid to average together nonstationary processes, like the majority of ring-width series, because such processes do not possess a defined mean or variance.”  Either way, I would start the x-axis at a rounder number, like 1550, or even 1500, and then increment more logically and more broadly in order to have way fewer axis labels.  I’d also get sample depths into this graph, though I see later that sample depths are in Figure 2.

Lines 213-214: “Both drawn and calculated TRW characteristics from all three localities despite small amplitudes showed rather high variation since 1850.”  How about a time series plot of running sta. dev.?  A classic way to depict changes in inter-annual variation through time.

Figure: OK, sample depths are included here.  Question: are these in terms of # of trees, or # of cores (not indicated in the axis labels)?  I prefer # of trees, as the tree is the basic unit of sampling in dendrochronology, not the core.

Line 280: “Red line is 20 years’ spline” should be “20-year spline.”

Line 282: WRI?  Should this be RWI?

Figure 3: Axis labels?

Figure 4: I would start the x-axis at 1900 and increment by 20 years.

Lines 409-411: “Mean monthly air temperature and monthly precipitation sums were retrieved from gridded climate data (CRU TS database show Fig. 5 A, B which covered the period 2019-2020.”  2019 is the start date of the climate data?  Surely this is a typo?  Figure 5 starts at 1901.

Figure 5: Speaking of this, I’d start it at 1900 and increment by 20.  Axis labels?

Line 497: “tree line uplift” should be “tree line migration.”  Uplift is not the correct word for this phenomenon.

Some other specific English fixes:

Line 18: “inconsistent respond” should be inconsistent response

Line 29: “what indicates” should be that indicates

Line 59: “this changes” should be these changes, or this change

Line 107: “most diffifultly accessible” should be “most inaccessible”

Line 138: “detrendization” should be detrending, or standardization

Line 155: “subactives” is not a word, at least not that I know of

Line 183: “all thee study sites” should be all three, or all the

Line 353: “inerpretation” should be interpretation

Line 602: “point years” should be pointer years

Plus, inconsistencies exist in the list of references.

Author Response

Dear reviewer,

Thank you very much for your comments and recommendations to improve our manuscript. Please find enclosed our replies.

In the Revised version 1.1 all the changes required and recommended by the Rev1, Rev2 and the English Editing Service are highlighted with yellow, green, and blue colours respectively. Final text is presented in the Revised version 1.2. Remote access to the office PC due to the restrictions did not allow to use “track changes” mode.

Round 2

Reviewer 2 Report

The article cited for COFECHA (used to be #38, now is #39) is still incorrect.  #39 is still the article about the ITRDB, not COFECHA.